# The Cocrystal of Ubiquinol: Improved Stability and Bioavailability

**DOI:** 10.3390/pharmaceutics15102499

**Published:** 2023-10-20

**Authors:** Qi Zhang, Mengyuan Xia, Chenxuan Zheng, Yinghong Yang, Junjie Bao, Wenjuan Dai, Xuefeng Mei

**Affiliations:** 1Pharmaceutical Analytical & Solid-State Chemistry Research Center, Shanghai Institute of Materia Medica, Chinese Academy of Sciences, Shanghai 201203, China; qizhang@simm.ac.cn (Q.Z.); xiamengyuan@simm.ac.cn (M.X.); zhengchenxuan@simm.ac.cn (C.Z.); yangyinghong@simm.ac.cn (Y.Y.); baojunjie@simm.ac.cn (J.B.); daiwenjuan@simm.ac.cn (W.D.); 2University of Chinese Academy of Sciences, No. 19A Yuquan Road, Beijing 100049, China; 3College of Pharmacy, Nanchang University, Nanchang 330006, China

**Keywords:** ubiquinol, cocrystal, polymorph, stability, bioavailability

## Abstract

Coenzyme Q10 (CoQ10) exists in two forms, an oxidized form and a reduced form. Ubiquinol is the fully reduced form of CoQ10. Compared to the oxidized form, ubiquinol has a much higher biological absorption and better therapeutic effect. However, ubiquinol has an important stability problem which hampers its storage and formulation. It can be easily transformed into its oxidized form—ubiquinone—even at low temperature. In this work, we designed, synthesized, and characterized a new cocrystal of ubiquinol with vitamin B3 nicotinamide (UQ-NC). Compared to the marketed ubiquinol form, the cocrystal exhibited an excellent stability, improved dissolution properties, and higher bioavailability. The cocrystal remained stable for a long period, even when stored under stressed conditions. In the dissolution experiments, the cocrystal generated 12.6 (in SIF) and 38.3 (in SGF) times greater maximum ubiquinol concentrations above that of the marketed form. In addition, in the PK studies, compared to the marketed form, the cocrystal exhibited a 2.2 times greater maximum total coenzyme Q10 concentration and a 4.5 times greater AUC than that of the marketed form.

## 1. Introduction

Coenzyme Q10 (CoQ10) is a vitamin-like, fat-soluble organic compound and it was identified for the first time by Frederick Crane of Wisconsin (USA) in 1957 [1,2]. It comprises a quinone group and a side-chain of 10 units of isoprene (Figure 1). It can be synthesized by human cells and is ubiquitously present in cell membranes [3,4]. CoQ10 plays a major role in supplying all cells with energy by shuttling electrons from complexes I and II to complex III of the mitochondrial respiratory chain [5,6,7,8,9,10]. It has been widely used in treating heart diseases and is marketed as a health supplement and for use in cosmetic products. However, the orally administered absorption efficiency of CoQ10 is very low (2–3%) due to its extremely low aqueous solubility, limited solubility in lipids, and relatively large molecular weight [11].

CoQ10 exists in three forms, the fully oxidized form, the radical semiquinone intermediate, and the fully reduced form [12]. Ubiquinol is the fully reduced form and ubiquinone is the fully oxidized form. Ubiquinol is the only lipophilic antioxidant that can be synthesized by human cells. It can neutralize free radicals, regenerate the reduced form of vitamin E [13], inhibit lipid peroxidation in biological membranes, and protect mitochondrial proteins and DNA from oxidative damage [14]. In contrast, ubiquinone requires enzymatic reduction before providing the above-mentioned functions. The enzymatic activities of coenzyme Q reductases change according to conditions, such as age, medications, and sports. In addition, compared to the oxidized form ubiquinone, ubiquinol has a much higher biological absorption and better therapeutic effect. Malkanthi Evans et al. [15] compared the bioavailability of a single, 100 mg/day dose of ubiquinol and ubiquinone in individuals of >60 years. Compared with the subjects receiving ubiquinone, the subjects receiving ubiquinol had a 4.3-fold higher plasma AUC_0–72h_ (430% increase). Ying Zhang et al. [16] compared the bioavailability of ubiquinol and ubiquinone in ten eligible older men randomized at a dose of 200 mg/day taken with one of the main meals. The significant increase in plasma CoQ10 status observed after the 2-week supplementation suggested that ubiquinol appeared to be a better supplemental form for enhancing the CoQ10 level as compared to ubiquinone. Fabio Marcheggiani [17] compared the anti-aging effects of ubiquinone and ubiquinol in a senescence model of human dermal fibroblasts. The results showed that ubiquinol was more efficient compared to ubiquinone in reverting the expression of the senescent phenotype, quantified in terms of β-galactosidase positivity, p21, collagen type 1, and elastin at the gene and protein expression levels. Severe-case heart failure and cardiac patients exhibit a remarkably poor CoQ10 absorption with low plasma CoQ10 levels. In a study of these patients in which they changed from taking ubiquinone to taking ubiquinol, the total CoQ10 content in their bodies increased, and the symptoms of heart failure were significantly improved [18,19]. In addition, ubiquinol demonstrates an excellent safety even at high doses. Recently, Jun Mitsui et al. [20] performed a multicenter, randomized, double-blinded, placebo-controlled phase 2 trial of ubiquinol in multiple-system atrophy. At a high dose of 1500 mg/day, ubiquinol was well-tolerated and led to a significantly smaller decline in UMSARS part 2 score compared to a placebo. However, ubiquinol has an important stability problem which hampers its storage and formulation [21]. It can be easily transformed into its oxidized form—ubiquinone—even at low temperature.

Several techniques have been used to improve the stability of ubiquinol. At present, the marketed form of ubiquinol is the polymorph II, which has been researched by the company Kaneka. Compared to the conventionally obtained form (polymorph I), the marketed form has a higher melting point, lower solubility, and better stability [22]. However, the stability experiment results show that the polymorph II demonstrates an insufficient stability over time. After four weeks, about 6.9% of ubiquinol in polymorph II was transformed to ubiquinone under open conditions at 25 °C. Recently, Rafel Prohens et al. [23] reported seven cocrystals of ubiquinol. Among them, the cocrystal of ubiquinol with 3,4-dihydroxybenzoic acid showed a remarkably high stability under standard stress conditions [24]. Regrettably, neither the patent nor the paper presents the specific content changes in the ubiquinol in those cocrystals during the stability experiments.

Nicotinamide, the metabolite of niacin in the body, is one form of vitamin B3. It is a food additive and can be found in many foods such as meat, fish, milk, eggs, green vegetables, and cereals. It is a potential means of preventing the onset of diabetes mellitus. An intervention study in New Zealand using nicotinamide treatment showed a 50% reduction in the development of insulin-dependent diabetes mellitus in a 5-year period [25]. In addition, the combination of niacin (the precursor of nicotinamide) and ubiquinone (the precursor of ubiquinol) can enhance therapeutic benefits in the treatment of diabetes [26]. Therefore, there may also be some synergies between niacinamide and ubiquinol in the treatment for diabetes.

In this work, we designed, synthesized and characterized a new cocrystal of ubiquinol with nicotinamide (UQ-NC). The cocrystal was fully characterized using PXRD, FTIR, and DSC. The stability experiments of the cocrystal were conducted under open conditions (25 °C, 60% relative humidity (RH), open) and accelerated conditions (40 °C, 75% RH, packaged with aluminum foil bags). In addition, the powder dissolution rate of the cocrystal was compared with the marketed form. Finally, the bioavailability of the cocrystal and the marketed form were compared in rats.

## 2. Materials and Methods

### 2.1. Materials

Ubiquinone was purchased from Zhejiang NHU Co., Ltd. (Shaoxing, China), with a greater than 98% purity. The conformer nicotinamide and the reductant ascorbic acid were purchased from Aladdin Co., Ltd., Shanghai, China. The other reagents and solvents were purchased from Sinopharm Chemical Reagent Co., Ltd., Shanghai, China. and used without further purification.

### 2.2. Preparation of Ubiquinol

In total, 10 g of ubiquinone and 6g of ascorbic acid were dissolved in 100 mL of ethanol at 80 °C protected by nitrogen. The solution was stirred for 24 h at 80 °C and then cooled to 0 °C slowly. White precipitate was obtained via filtration and the filter cake was washed three times with ethanol. Finally, ubiquinol was obtained via vacuum drying as a white powder. The purity of the ubiquinol was determined to 98.6% using HPLC and the residual ascorbic acid was controlled to less than 0.1%. ^1^H NMR (CDCl3, Appendix A) δ 5.32 (s, 1H), 5.29 (s, 1H), 5.11 (dd, J = 7.6, 4.5 Hz, 8H), 5.10 (s, 1H), 3.89 (s, 6H), 3.33 (d, J = 6.6 Hz, 2H), 2.14 (s, 3H), 2.08–2.04 (m, 18H), 1.99 -1.96 (m, 18H), 1.77 (d, J = 0.7 Hz, 3H), 1.68 (d, J = 1.0 Hz, 3H), and 1.60 (s, 27H).

### 2.3. Preparation of Polymorph II of Ubiquinol^22^

A total of 40 g of ubiquinol was dissolved in 90 mL of hexane at 40 °C protected by nitrogen and then cooled to 25 °C at a cooling rate of 10 °C/h. The mixture was stirred for 96 h at 25 °C. The temperature was strictly controlled. If the temperature was too high, the crystal could not be precipitated; if the temperature was too low, the crystal would precipitate too quickly to obtain the polymorph II. Finally, the polymorph II of the ubiquinol was obtained via filtration and vacuum drying as a light-yellow powder.

### 2.4. Preparation of the Cocrystal UQ-NC

In total, 0.244 g of nicotinamide, 1.72 g of ubiquinol, and 1 mL of a solvent of isopropanol/isopropyl acetate in a 1:1 ratio were added to a ball mill jar. The ball milling was performed using a jingxin JX-2013 ball miller for 2 h. The mixture was dried in a vacuum oven at room temperature for 12 h to render the cocrystal UQ-NC as a white powder.

### 2.5. Powder X-ray Diffraction (PXRD)

A Bruker D8 Advance X-ray diffractometer (Cu Kα radiation) was used to collect the PXRD patterns. The voltage was set to 40 kV and current was set to 40 mA, respectively. In the process of the experiments, samples were placed on the sample holder and then scanned from 3 to 40° 2θ in reflection mode with a scan rate of 15°/min (step size 0.025°, step time 0.1 s). The data were imaged and integrated with RINT Rapid and peak-analyzed with Jade 6.0 from Rigaku, Woodlands, TX, USA.

### 2.6. Differential Scanning Calorimetry (DSC)

DSC experiments were performed using a TA Q2000 instrument under nitrogen gas with a flow rate of 20 mL/min. Approximately 3–5 mg of ground sample was put into an aluminum pan and covered with a lid. Then, the aluminum pan was heated from 20 to 140 °C with a heating rate of 10 K/min. The instrument was calibrated using indium and tin to check the temperature axis and heat flow.

### 2.7. Fourier Transformation Infrared (FTIR) Spectroscopy

FTIR spectra were collected in the range from 4000 to 350 cm^−1^ at ambient conditions using a Nicolet-Magna FT-IR 750 spectrometer manufactured by Thermo Fisher Scientific Co., Ltd., Waltham, MA USA. The resolution was 4 cm^−1^.

### 2.8. Stability Study

Open stability was conducted at 25 °C, 60% RH, under open conditions without package and nitrogen protection. Accelerated stability was conducted at 40 °C, 75% RH, packaged with aluminum foil bags. The ubiquinol polymorph II and UQ-NC cocrystal were taken out at various intervals during exposure (open stability at 0, 3, 7, 14, 21, and 28 days and accelerated stability at 0, 1, 2, 3, and 6 months).

### 2.9. Powder Dissolution

Powder dissolution experiments were conducted in simulated intestinal fluid (SIF) and simulated gastric fluid (SGF) with 0.5% tween 80 at 37 °C using a Mini-Bath dissolution device equipped with a Julabo ED-5 heater/circulator. SIF was prepared by dissolving 6.8 g of KH_2_PO_4_ in 500 mL of water and then the pH was adjusted to 6.8 with 0.1 mol/L og NaOH solution. In total, 10 g of pancreatin was added and finally the solution was diluted to 1000 mL with water to obtain the SIF. SGF was prepared by mixing 16.4 mL of 10% HCl solution, 800 mL of water, and 10g of pepsin. Finally, the solution was diluted to 1000 mL with water to obtain the SGF. A total of 9.3 mg of cocrystal and 8.2 mg of polymorph II were added into 20 mL of the buffer solution. The stirring speed was set to 75 rpm. To minimize the size effect on the dissolution results, the ubiquinol cocrystal and polymorph II were sieved through 80-mesh sieves. Samples were collected at 5, 15, 20, 25, 30, 40, 50, 60, 80, 130, and 180 min each time, and were filtered through the 0.45 μm nylon filters. The concentration of ubiquinol was determined using HPLC. The detailed HPLC analysis method is described in Section 2.10.

### 2.10. High Performance Liquid Chromatography (HPLC) Analysis

An Agilent 1260 series HPLC was used to determine the content of ubiquinol. The HPLC was equipped with a quaternary pump (G1311C), diode-array detector (G1315D), and a 4.6 mm × 100 mm, 2.7 μm packing L1 column. The detector was set at 290 nm. A mobile phase consisting of methanol and n-hexane (85/15, *v*/*v*) was run for 10 min at 1.5 mL/min. An injection of 20 μL was performed and the column temperature was set at 35 °C.

### 2.11. Pharmacokinetic Study

Pharmacokinetic experiments were conducted on male Sprague Dawley rats to compare the oral exposure of the UQ-NC cocrystal and the ubiquinol marketed form II. The two ubiquinol samples were dispersed homogeneously in 0.5% CMC-Na (sodium carboxymethylcellulose) and 0.5% tween 80 aqueous solution to obtain the suspension of 10 mg/mL of ubiquinol. Twelve male Sprague Dawley rats weighing 220–250 mg were randomly allocated into two groups (six rats in each group). Each received gavage administration at a dose of 100 mg/kg body (expressed as ubiquinol equivalents). After this administration, approximately 200 μL of blood sample was collected from the orbital sinus into heparinized tubes at 0, 1, 2, 3, 4, 6, 8, and 24 h for each rat.

In this experiment, the whole process was protected from light. An aliquot of 100 µL of plasma sample was added to 100 µL of p-benzoquinone and then placed for 10 min after vortex mixing. Next, a 400 µL methanol solution containing 1 µg/mL of IS Q9 and 1mL of n-hexane was added. The mixture was vortexed for 5 min and centrifuged at 14,000 rpm for 1 min. In total, 0.825 mL of the upper extract was transferred and dried under the condition of 30 °C nitrogen, and 200 µL of ethanol equal solution was added for re dissolution, vortexed, and mixed well. An aliquot of 4.00 µL of supernatant was injected for an LC-MS/MS analysis.

### 2.12. Bioanalytical Method

The quantification of coenzyme Q10 in the plasma samples was accomplished using an AB Sciex Triple Quad 4500 LC-MS instrument manufactured by SCIEX Co., Ltd., Framingham, MA, USA. Chromatography consisted of a Column NO.ZORBAX Eclipse Plus C18 (50 mm × 4.6 mm, 1.8 μm). The mobile phase consisted of acetonitrile and alcohol (50/50, *v*/*v*) at a flow rate of 1.0 mL min-1. An injection of 10 μL was performed and the column temperature was set to 40 °C. The detector was set at 290 nm. Multiple reaction monitoring modes (MRM) were used to perform the detection and quantification, with *m*/*z* 863.8→197.4 for coenzyme Q10. The DAS 2.0 program was used to perform the PK analysis.

## 3. Results

### 3.1. Preparation and PXRD Analysis for Different Crystalline Forms

Two polymorphs of ubiquinol have been reported. Polymorph I is a conventionally obtained crystalline form. Polymorph II is synthesized by cooling and slurrying. The UQ-NC cocrystal is synthesized via ball milling. The contents of ubiquinol and nicotinamide in the cocrystal were measured, respectively, using HPLC, and it was found that the content of ubiquinol was about 86.1%, and the content of nicotinamide was about 12.6%, indicating that, in this co-crystal, the stoichiometric ratio of ubiquinol to nicotinamide was about 1:1. The stoichiometric ratio was also verified using ^1^H NMR (Appendix A). In the ^1^H NMR spectrum, the peak areas (normalized) of the peaks at δ 9.02, 8.77, 8.17, and 7.42 corresponding to the hydrogen atoms on the pyridine ring of the nicotinamide were 1.0, 0.99, 0.97, and 0.99, respectively. Each peak represents one hydrogen atom. The peak area (normalized) of the peak at δ 3.88 corresponding to the hydrogen atoms on the methoxy groups of the ubiquinol was 6.37. This peak represents six hydrogen atoms. Such results indicated that the stoichiometric ratio of ubiquinol to nicotinamide was about 1:1.

PXRD is primarily used to confirm cocrystal formation, wherein different characteristic peaks appear. The PXRD patterns of the three different crystalline forms of ubiquinol are presented in Figure 1. The polymorph I presents characteristic peaks at 2θ 2.9°, 4.6°, 6.1°, and 9.2° and the polymorph II presents characteristic peaks at 2θ 6.4°, 10.8°, 12.9°, and 17.6°. In comparison, the UQ-NC cocrystal exhibits new unique peaks at 2θ 5.6°, 9.9°, 12.7°, 19.8°, and 20.8°. The clearly different PXRD patterns suggests the generation of a new phase—the UQ-NC cocrystal.

### 3.2. FTIR Spectra

IR spectra were further utilized to confirm the cocrystal generation and shed some light on the driving force behind the stabilization. The IR spectra of ubiquinol, nicotinamide, and the cocrystal are presented in Figure 2 (the polymorphs I and II exhibit similar IR spectra, here only presents the IR spectrum of polymorph I). The results show that the IR spectrum of the cocrystal is different from that of the two raw materials. The differences mainly appeared in the wavenumber range between 3000 and 4000 cm^−1^. In the IR spectrum of ubiquinol, there is a broad peak at 3453.9–3533.0 cm^−1^ representing the O-H stretching in the ubiquinol molecule. In the IR spectrum of nicotinamide, there is a sharp peak at 3365.2 cm^−1^ representing the N-H stretching in the nicotinamide molecule. In the UQ-NC cocrystal, the two peaks were overlapped and there is a sharp peak at 3465.5 cm^−1^ representing the N-H stretching in the nicotinamide molecule. Compared to the peak in the IR spectrum of nicotinamide, there is a 100 cm^−1^ blue shift. This blue shift indicates the strong hydrogen bond interaction between ubiquinol and nicotinamide.

### 3.3. DSC Analysis

The overlaid DSC curves of ubiquinol’s different crystalline forms are presented in Figure 3. The results show that polymorph I has an onset value at 48.4 °C and a peak point at 49.5 °C, polymorph II exhibits an onset value at 51.6 °C and a peak point at 54.0 °C, and the cocrystal shows an onset value at 57.8 °C and a peak point at 59.7 °C, respectively. The cocrystal has a higher melting point than that of the two polymorphs. The much higher melting point of the QU-NC cocrystal may be attributed to its stronger intermolecular interaction, which is also reflected in the greater blueshift of the O-H stretching in the IR spectra. The higher melting point may also be one of the reasons for the improved stability of the UQ-NC cocrystal.

### 3.4. Stability Studies

The open stability of ubiquinol polymorph I, polymorph II, and the UQ-NC cocrystal were investigated at 25 °C, 60% RH, under open conditions. The results (Figure 4) showed that, after 4 weeks, the assay values of ubiquinol were decreased to 28.3% and 94.2% for polymorph I and II, respectively. In comparison, the ubiquinol content in the UQ-NC cocrystal only decreased to 97.8%. The cocrystal retained this stability for an extended period without deliberately taking protective measures against oxygen, indicating that the ubiquinol cocrystal of the present invention had a significantly improved stability compared to the ubiquinol itself. Furthermore, the cocrystal was subjected to an accelerated stability investigation. Three batches of the UQ-NC cocrystal samples were packaged in aluminum foil bags and then stored under 40 °C, 75% RH conditions. The samples were taken out at various intervals and the content of ubiquinol was determined at 1, 2, 3, and 6 months. The results are presented in Appendix A and Table 1. After storage under 40 °C, 75% RH conditions for six months, the PXRD patterns of the cocrystal remained unchanged and the ubiquinol contents in the three batches of the UQ-NC cocrystal were 98.9%, 98.1%, and 98.2%, respectively. Such results showed that the cocrystal could remain stable for an extended period even when stored under stressed conditions. Through co-crystallization technology, the stability of ubiquinol was successfully improved. There are two reasons which may have contributed to the cocrystal stability improvement. Firstly, the oxygen penetration of the interior of the crystal was prevented by compact crystallization; secondly, the reactive phenolic hydroxy groups were protected by the hydrogen bonds between the phenol hydroxyl and pyridine nitrogen groups.

### 3.5. Powder Dissolution Study

Dissolution experiments of the UQ-NC cocrystal and ubiquinol marketed form II were conducted in SGF and SIF with 0.5% tween 80 at 37 °C. To minimize the size effect on the dissolution results, the ubiquinol cocrystal and polymorph II were sieved through 80-mesh sieves. After the dissolution experiments, the residual solid samples were checked using PXRD and the crystalline forms were unchanged (Appendix A). The dissolution data are presented in Figure 5 and summarized in Table 2. The results showed that the cocrystal had superior dissolution properties compared to the marketed form. The cocrystal generated 12.6 (SIF) and 38.3 (SGF) times greater maximum ubiquinol concentrations above that of the marketed form II, and the concentrations of ubiquinol in the cocrystal were still growing after 3 h. Area under the curve (AUC) values are representative of the exposure of the active pharmaceutical ingredient (API). For cocrystals, AUCs are also directly proportional to dissolution and inversely proportional to precipitation. In this work, the AUCs of the cocrystal were 29.6 (SIF) and 76.3 (SGF) times that of the marketed form II.

### 3.6. In Vivo Pharmacokinetic Study

After satisfactory in vitro dissolution studies, in vivo pharmacokinetic investigations of the UQ-NC cocrystal and the marketed form II were carried out on fasted male rats. The pharmacokinetic parameters are shown in Table 3 and the mean plasma concentrations of the total coenzyme Q10 versus the time profiles are presented in Figure 6. The results show that the UQ-NC cocrystal had a much higher in vivo bioavailability than that of the marketed form II. Compared to the marketed form II, the cocrystal had a shorter T_1/2_, T_max_, and MRT, indicating that the cocrystal was absorbed much faster. The C_max_ of the cocrystal and the marketed form II were 157.7 ng/mL and 71.2 ng/mL, respectively. The cocrystal generated 2.2 times maximum total coenzyme Q10 concentrations above that of the marketed form II. In addition, the plasma exposures (AUC_0–8h_) of the cocrystal and the marketed form II were 866.5 h·ng/mL and 191.9 h·ng/mL, respectively. The cocrystal generated a 4.5 times greater AUC than that of the marketed form II. The PK studies demonstrated that the oral absorption of ubiquinol can indeed be enhanced by its cocrystal, originating from the improved solubility and dissolution rate.

## 4. Discussion

In order to avoid the influence of impurities on the stability study, the same batch of ubiquinol raw material was used to prepare the cocrystal and polymorph II of ubiquinol. The cocrystal and polymorph II were determined using HPLC. Except for nicotinamide, they had similar impurities. In addition, the residual ascorbic acid in the ubiquinol raw material was controlled to less than 0.1%. Hence, the improvement in stability was not caused by impurities. The reduced lattice energy should account for the increased stability, and the increased melting point can confirm the change in the lattice energy.

According to Yalkowsky and Valvani [27], the aqueous solubility, Xa, is generally defined as: log Xa = log Xi − log γ. Xi is the ideal solubility and γ is the activity coefficient of the solute in water. The ideal solubility Xi is exclusively dependent on the melting point (representative of the crystal lattice energy) of the solute and is the same for every solvent media used. The ideal solubility of a solute can be mathematically expressed as: log Xi = −0.01(Tm-25). Tm represents the melting point of the solute in degrees Celsius. A higher melting point always leads to a lower Xi. The activity coefficient term represents the decrease in the aqueous solubility due to intermolecular interaction differences in solute and solvent molecules (representative of the solvation energy). In other words, the activity coefficient γ can be considered as the measurement of the intermolecular forces of attraction to be overcome, for removing a molecule from the solute and placing it in the solvent. Higher melting points may lead to a lower Xi, thus reducing the solubility. However, the introduction of nicotinamide can increase the intermolecular interaction between the solute and the dissolving medium, then improving the solubility. The combined effect of the two influencing factors causes a solubility improvement in the cocrystal. For form II, ubiquinol interacts with H_2_PO_4_^−^ via strong hydrogen bonds in SIF, but not in SGF. Hence, in SIF, the ubiquinol molecule is more easily placed into the solvent and the solubility of form II is higher in SIF than in SGF. In the cocrystal, the ubiquinol molecule was hydrogen bonded by nicotinamide and the influence of H_2_PO_4_^−^ was limited. Hence, the solubility of the cocrystal was similar in SIF and SGF.

The ubiquinol absorption process can be divided into three parts. The first step is ubiquinol’s arrival in the intestinal lumen. In this step, the stability is a key influencing factor. The more stable the crystalline forms are, the more ubiquinol reaches the intestine. The enhanced stability of the cocrystal may be one reason for its increased bioavailability. The second step is the dissolution of ubiquinol, and the higher solubility of the cocrystal is another factor improving absorption. In the third step, the dissolved ubiquinol enters the enterocytes via simple passive facilitated diffusion. Finally, ubiquinol is incorporated into the chylomicrons and subsequently reaches the bloodstream through the lymphatic system. In the last two steps, the ubiquinol has been dissolved and the process is independent of the solid forms. In summary, the enhanced stability and increased solubility may contribute to the improved bioavailability.

## 5. Conclusions

In conclusion, we successfully improved the stability and bioavailability of ubiquinol via hydrogen-bond driven co-crystallization. A new cocrystal of ubiquinol with vitamin B3 nicotinamide (UQ-NC) was prepared. The cocrystal was fully characterized using PXRD, FTIR, Raman, and DSC. Stability experiments of the cocrystal were conducted and the cocrystal presented an excellent stability. In addition, the powder dissolution rate of the cocrystal was compared with the marketed form. The cocrystal generated 12.6 (in SIF) and 38.3 (in SGF) times greater maximum ubiquinol concentrations than that of the marketed form II. Finally, the bioavailability of the cocrystal and the marketed form were compared in rats. The results showed that the cocrystal exhibited 2.2 times greater maximum total coenzyme Q10 concentrations and a 4.5 times greater AUC than that of the marketed form.

## Data Availability

No new data were created.

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
