# Peer review of "The Cocrystal of Ubiquinol: Improved Stability and Bioavailability"

_pharmaceutics, 2023, doi:10.3390/pharmaceutics15102499_

Round 1

Reviewer 1 Report

Manuscript (Cocrystal of ubiquinol: improved stability and bioavailability) aimed to evaluate a new cocrystal of ubiquinol with vitamin B3 nicotinamide (UQ-NC) and compare it with the marketed form to improve the stability and dissolution properties and enhance the bioavailability of ubiquinol. The work has shown improvement in clinical and biochemical parameters that were statistically significant. The experimental work is well-designed, the results are promising, and the discussion is convenient and logical. However, some comments and inquiries need to be issued:

1-     Linguistic revision should be maintained for the manuscript.

2-    The abstract needs some modification to be fully representative of the obtained results, It only gives an idea about the objectives of the work with little idea about the methodology and the obtained results.

3-    In section 2.7: (methodology) the subtitle was “FTIR” and the results section 3.2: the subtitle was “vibrational spectra”, I suggest unifying the subtitle.

4-    Also, please unify the subtitle “DSC” in methodology and “thermal analysis” in Results.

5-    Table 1: Please revise its content (regarding the name of the compared formulas)

6-    Table 2: It is better to say Pharmacokinetic parameters in the caption instead of kinetic parameters.

7-    Why the researchers didn’t include other pharmacokinetic parameters such as Tmax, MRT, and elimination t1/2, I advise the authors to include them and discuss them.

8-    I suggest future work regarding the use of this cocrystal of ubiquinol and niacinamide and their expected synergistic effect for the treatment of Diabetes.

Linguistic revision should be maintained for the manuscript.

Author Response

omments and Suggestions for Authors

Manuscript (Cocrystal of ubiquinol: improved stability and bioavailability) aimed to evaluate a new cocrystal of ubiquinol with vitamin B3 nicotinamide (UQ-NC) and compare it with the marketed form to improve the stability and dissolution properties and enhance the bioavailability of ubiquinol. The work has shown improvement in clinical and biochemical parameters that were statistically significant. The experimental work is well-designed, the results are promising, and the discussion is convenient and logical. However, some comments and inquiries need to be issued:

  • Linguistic revision should be maintained for the manuscript.

Response: The language had been revised carefully.

  • The abstract needs some modification to be fully representative of the obtained results, It only gives an idea about the objectives of the work with little idea about the methodology and the obtained results.

Response: Revised as suggested.

  • In section 2.7: (methodology) the subtitle was “FTIR” and the results section 3.2: the subtitle was “vibrational spectra”, I suggest unifying the subtitle.

Response: Revised as suggested.

  • Also, please unify the subtitle “DSC” in methodology and “thermal analysis” in Results.

Response: Revised as suggested.

  • Table 1: Please revise its content (regarding the name of the compared formulas)

Response: Revised as suggested.

Table 2: It is better to say Pharmacokinetic parameters in the caption instead of kinetic parameters.

Response: Revised as suggested.

6-    Why the researchers didn’t include other pharmacokinetic parameters such as Tmax, MRT, and elimination t1/2, I advise the authors to include them and discuss them.

 Response: Revised as suggested.

7-    I suggest future work regarding the use of this cocrystal of ubiquinol and niacinamide and their expected synergistic effect for the treatment of Diabetes.

Response: Thanks for suggestion. The expected synergistic effect for the treatment of Diabetes is in processing.

Reviewer 2 Report

The manuscript under review presents the synthesis, characterization and pharmaceutical assessment of a nicotinamide:ubiquinol cocrystal.  While this is a new cocrystal formulation of ubiquinol, it shares similar properties as a previously published cocrystal with dihydroxybenzoic acid.  This particular formulation is found to have enhanced solubility and stability properties, which are supported by the reported studies.

  1. The authors included duplicate NMR spectra of the cocrystal product in the Supporting Information.  Please include the oringinal Ubiquinol NMR spectrum in Figure S1.
  2. The authors should provide evidence of the stoichiometry of the cocrystal from the NMR spectrum.  Please provide integral values that show the 1:1 complex as claimed.
  3. Lines 87-88, Raman is included in listed characterization, but no experimental data is provided.  Either provide the data or remove the reference.
  4. The melting point of the cocrystal is higher than either polymorphs of ubiquinol.  Generally this would indicate a more stable crystalline form, and a correspondingly lower solubility. Can the authors explain why this is not the case in this specific circumstance?
  5. The OH stretch of ubiquinol is claimed to shift to higher frequency upon formation of the cocrystal.  This peak is significantly sharper in the cocrystal form and similar to the sharp  Nicotinamide has a similar sharp NH asymmetric stretch in this region as shown in Figure 2.   Is it possible that this is not evidence of hydrogen bonding, but simply the NH stretch of nicotinamide?
  6. The experimental description of the dissolution experiments require more detail, including the exact compositions of the buffers used and dose of the active substance, as well as details of the HPLC method used for analysis.
  7. Can the authors explain why the solubility of form II is three fold higher in SIF over SGF?  Is this attributable to the presence of surfactant in the SIF?  If so, why is a similar effect not observed for the cocrystal?
  8. The interpretation of the in-vivo PK data are lacking mechanistic interpretation.  Is the enhanced solubility of ubiquinol solely responsible for the PK profile observed? What is the role of nicotinamide absorption?  What is responsible for the decrease in plasma concentration of the cocrystal to match that of Form II at longer times?
  9.  Is the increase in stability observed solely due to resistance to oxidation?  Was the impurity profile and levels of related substances observed in the cocrystal the same as Form II? 

Use of an english language editor is recommended to fix several grammatical errors noted throughout the manuscript.

Author Response

The manuscript under review presents the synthesis, characterization and pharmaceutical assessment of a nicotinamide:ubiquinol cocrystal.  While this is a new cocrystal formulation of ubiquinol, it shares similar properties as a previously published cocrystal with dihydroxybenzoic acid.  This particular formulation is found to have enhanced solubility and stability properties, which are supported by the reported studies.

  1. The authors included duplicate NMR spectra of the cocrystal product in the Supporting Information.  Please include the oringinal Ubiquinol NMR spectrum in Figure S1.

Response: Revised as suggested.

  1. The authors should provide evidence of the stoichiometry of the cocrystal from the NMR spectrum.  Please provide integral values that show the 1:1 complex as claimed.

Response: Revised as suggested.

  1. Lines 87-88, Raman is included in listed characterization, but no experimental data is provided.  Either provide the data or remove the reference.

Response: Revised as suggested.

  1. The melting point of the cocrystal is higher than either polymorphs of ubiquinol.  Generally this would indicate a more stable crystalline form, and a correspondingly lower solubility. Can the authors explain why this is not the case in this specific circumstance?

Response: According to Yalkowsky and Valvani1, the aqueous solubility, Xa, is generally defined as: log Xa = log Xi - log g. Xi is the ideal solubility and g is the activity coefficient of the solute in water. The ideal solubility is exclusively dependent on the melting point (representative of crystal lattice energy) of solute and is the same for every solvent media used. The ideal solubility of a solute can be mathematically expressed as: log Xi = -0.01(Tm-25). Tm represents the melting point of solute in Celsius. The activity coefficient term represents the decrease in aqueous solubility due to intermolecular interaction differences of solute and solvent molecules (representative of solvation energy). In other words, activity coefficient g can be considered as the measurement of the intermolecular forces of attraction to be overcome, for removing a molecule from the solute and placing it in the solvent. Higher melting points may lead to lower Xi, thus reducing the solubility. However, introduction of nicotinamide can increase the intermolecular interaction between the solute and the dissolving medium and then improve the solubility. The combined effect of the two influencing factors causes the solubility improvement.

  1. The OH stretch of ubiquinol is claimed to shift to higher frequency upon formation of the cocrystal.  This peak is significantly sharper in the cocrystal form and similar to the sharp  Nicotinamide has a similar sharp NH asymmetric stretch in this region as shown in Figure 2.   Is it possible that this is not evidence of hydrogen bonding, but simply the NH stretch of nicotinamide?

Response: yes, the sharp peak represents the N-H stretching in nicotinamide molecule. Compared to the peak in the IR spectrum of nicotinamide, there is a 100 cm-1 blue shift. This blue shift indicates the strong hydrogen-bond interaction between ubiquinol and nicotinamide.

  1. The experimental description of the dissolution experiments require more detail, including the exact compositions of the buffers used and dose of the active substance, as well as details of the HPLC method used for analysis.

Response: Revised as suggested.

  1. Can the authors explain why the solubility of form II is three fold higher in SIF over SGF?  Is this attributable to the presence of surfactant in the SIF?  If so, why is a similar effect not observed for the cocrystal?

Response: According to Yalkowsky and Valvani1, the aqueous solubility, Xa, is generally defined as: log Xa = log Xi - log g. Xi is the ideal solubility and g is the activity coefficient of the solute in water. The ideal solubility is exclusively dependent on the melting point (representative of crystal lattice energy) of solute and is the same for every solvent media used. Hence, the solubility difference of form II in SIF and SGF should be affected by the activity coefficient g. The activity coefficient term represents the decrease in aqueous solubility due to intermolecular interaction differences of

solute and solvent molecules. In other words, activity coefficient g can be considered as the measurement of the intermolecular forces of attraction to be overcome, for removing a molecule from the solute and placing it in the solvent. For form II, ubiquinol interacts with H2PO4- by strong hydrogen bonds in SIF, but not in SGF. Hence, in SIF, the ubiquinol molecule is more easily to be placed into the solvent and the solubility of form II is higher in SIF over in SGF. In the cocrystal, the ubiquinol molecule is hydrogen bonded by nicotinamide and the influence of H2PO4- was limited.

  1. The interpretation of the in-vivo PK data are lacking mechanistic interpretation.  Is the enhanced solubility of ubiquinol solely responsible for the PK profile observed? What is the role of nicotinamide absorption?  What is responsible for the decrease in plasma concentration of the cocrystal to match that of Form II at longer times?

Response: The ubiquinol absorption process can be divided into three parts2. The first step is ubiquinol’s arrival in intestinal lumen. In this step, the stability is a key influencing factor. The more stable the crystalline forms are, the more ubiquinol reaches the intestine. The enhanced stability of the cocrystal may be one reason for its increased bioavailability. The second step is the dissolution of ubiquinol, and the higher solubility of the cocrystal is another factor improving absorption. In the third step, the dissolved ubiquinol enter the enterocytes via the simple passive facilitated diffusion. Finally, ubiquinol is incorporated in the chylomicrons and subsequently reaches the bloodstream through the lymphatic system. In the last two steps, the ubiquinol has been dissolved and the process is independent of the solid forms. In summary, the enhanced stability and the increased solubility may contribute to the improved bioavailability.

There is no evidence that nicotinamide is working in the ubiquinol absorption process。

The decrease in plasma concentration of the cocrystal to match that of Form II at longer times may be due to the difference in the absorption rate of ubiquinol.

  1.  Is the increase in stability observed solely due to resistance to oxidation?  Was the impurity profile and levels of related substances observed in the cocrystal the same as Form II? 

Response: In order to avoid the influence of impurities on the stability study, the same batch of ubiquinol raw material is used to prepare the cocrystal and polymorph II of ubiquinol. The cocrystal and polymorph II were determined by HPLC. Except for nicotinamide, they have similar impurities. In addition, the residual ascorbic acid in the ubiquinol raw material was controlled to less than 0.1%. Hence, the improvement in stability is not caused by impurities. The reduced lattice energy should account for the increased stability and the increased melting point can confirm the change of the lattice energy.

Comments on the Quality of English Language

Use of an english language editor is recommended to fix several grammatical errors noted throughout the manuscript.

Response: The language had been revised carefully.

Reference:

1.Shashank Jain, Niketkumar Patel, and Senshang Lin. Solubility and dissolution enhancement strategies: current under-standing and recent trends. Drug Dev Ind Pharm, 2014, 41:6, 875-887.

2.Alma Martelli , Lara Testai, Alessandro Colletti and Arrigo F. G. Cicero. Coenzyme Q10: Clinical Applications in Cardio-vascular Diseases. Antioxidants, 2020, 9, 341.

Reviewer 3 Report

In this study, the authors showed that the cocrystal of ubiquinol/nicotinamide improved the stability and solubility of ubiquinol. This report is expected to provide a novel pharmaceutical technology for medical application of ubiquinol. Although the draft is overall well written, some revisions should be addressed before publication as listed below.

1. In the synthesis of ubiquinol, the purity should be described. Impurities such as ascorbic acid can affect the result. If preparative HPLC was performed, describe some conditions related to it.

2. Overall, method overlap should be removed in the results section.

3. In general, the number of the figure concerned should be given in the explanation of the results.

4. In dissolution test, describe how much powder was used in how much solution. And some stirring condition (RPM) should also be required.

5. In pharmacokinetic study, is it acceptable to co-administer Tween80?

6. Overall, discussion section is needed. Or revise as "Results and Discussion". And discussion such as interpretation of data, comparison with previous reports and future aspect should be described.

Author Response

In this study, the authors showed that the cocrystal of ubiquinol/nicotinamide improved the stability and solubility of ubiquinol. This report is expected to provide a novel pharmaceutical technology for medical application of ubiquinol. Although the draft is overall well written, some revisions should be addressed before publication as listed below.

  1. In the synthesis of ubiquinol, the purity should be described. Impurities such as ascorbic acid can affect the result. If preparative HPLC was performed, describe some conditions related to it.

Response: The purity of ubiquinol was determined to 98.6% by HPLC and the residual ascorbic acid was controlled to less than 0.1%. Preparative HPLC was not performed

  1. Overall, method overlap should be removed in the results section.

Response: Revised as suggested.

  1. In general, the number of the figure concerned should be given in the explanation of the results.

Response: Revised as suggested.

  1. In dissolution test, describe how much powder was used in how much solution. And some stirring condition (RPM) should also be required.

Response: Revised as suggested.

  1. In pharmacokinetic study, is it acceptable to co-administer Tween80?

Response: it is acceptable of ubiquinol to co-administer tween80, but tween 80 could not improve the bioavailability of ubiquinol.

  1. Overall, discussion section is needed. Or revise as "Results and Discussion". And discussion such as interpretation of data, comparison with previous reports and future aspect should be described.

Response: Discussion section is added accordingly.
